# *MUTYH* as an Emerging Predictive Biomarker in Ovarian Cancer

**DOI:** 10.3390/diagnostics11010084

**Published:** 2021-01-06

**Authors:** Megan L. Hutchcraft, Holly H. Gallion, Jill M. Kolesar

**Affiliations:** 1Division of Gynecologic Oncology, Department of Obstetrics & Gynecology, University of Kentucky Markey Cancer Center, 800 Rose Street, Lexington, KY 40536-0263, USA; megan.hutchcraft@uky.edu (M.L.H.); Holly.Gallion1@uky.edu (H.H.G.); 2Department of Pharmacy Practice & Science, University of Kentucky College of Pharmacy, 567 Todd Building, 789 South Limestone Street, Lexington, KY 40539-0596, USA

**Keywords:** *MUTYH*, ovarian cancer, predictive biomarker, base excision repair

## Abstract

Approximately 18% of ovarian cancers have an underlying genetic predisposition and many of the genetic alterations have become intervention and therapy targets. Although mutations in MutY homolog (*MUTYH*) are best known for *MUTYH* associated polyposis and colorectal cancer, it plays a role in the development of ovarian cancer. In this review, we discuss the function of the *MUTYH* gene, mutation epidemiology, and its mechanism for carcinogenesis. We additionally examine its emerging role in the development of ovarian cancer and how it may be used as a predictive and targetable biomarker. *MUTYH* mutations may confer the risk of ovarian cancer by the failure of its well-known base excision repair mechanism or by failure to induce cell death. Biallelic germline *MUTYH* mutations confer a 14% risk of ovarian cancer by age 70. A monoallelic germline mutation in conjunction with a somatic *MUTYH* mutation may also contribute to the development of ovarian cancer. Resistance to platinum-based chemotherapeutic agents may be seen in tumors with monoallelic mutations, but platinum sensitivity in the biallelic setting. As *MUTYH* is intimately associated with targetable molecular partners, therapeutic options for *MUTYH* driven ovarian cancers include programed-death 1/programed-death ligand-1 inhibitors and poly-adenosine diphosphate ribose polymerase inhibitors. Understanding the function of *MUTYH* and its associated partners is critical for determining screening, risk reduction, and therapeutic approaches for *MUTYH*-driven ovarian cancers.

## 1. Introduction

Ovarian cancer, which generally refers to cancers originating from the ovary, fallopian tube, and peritoneum, is the deadliest gynecologic cancer, responsible for taking the lives of approximately 14,000 women in the United States this year [1]. Although most ovarian cancers are sporadic, approximately 18% have an underlying genetic predisposition [2]. The best known and most penetrant germline genetic mutations responsible for ovarian cancer are in the tumor suppressor breast cancer susceptibility genes 1 and 2 (*BRCA1* and *BRCA2*), which are noted in 13–20% of ovarian cancers [3,4]. These genes help maintain genome integrity by performing deoxyribonucleic acid (DNA) repair via the homologous recombination repair pathway [5]. *BRCA1* and *BRCA2* (*BRCA1*/*2*) mutations are inherited in an autosomal dominant fashion and contribute to an increased risk of several cancers including breast, ovarian, and pancreatic cancers [6]. *BRCA1* carriers are estimated to have a 44% risk of developing ovarian cancer and a 72% risk of developing breast cancer by age 80 [7]. Similarly, *BRCA2* carriers incur a 17% risk of developing ovarian cancer and a 69% risk of developing breast cancer by age 80 [7]. Evaluation of families with multiple cases of breast and/or ovarian cancer have demonstrated an approximate 40% mutation frequency in the *BRCA1*/*2* genes [8], suggesting less prevalent germline mutations contribute to the development of hereditary breast and ovarian cancer.

A germline mutation in the tumor protein 53 (*TP53*) tumor suppressor gene results in the rare, autosomal dominantly inherited Li Fraumeni hereditary cancer syndrome. Although Li Fraumeni syndrome has not classically been associated with an increased risk for ovarian cancer, higher than expected frequencies of ovarian cancer in patients with germline TP53 mutations have been reported [9]. Other gene mutations that confer ovarian cancer risk include those involved in the development of Lynch syndrome, or hereditary non-polyposis colorectal cancer. These include mutations in the mismatch repair genes mutL homolog 1 (*MLH1*), mutS homologs 2 and 6 (*MSH2*, *MSH6*), post-meiotic segregation increased 2 (*PMS2*), and epithelial cell adhesion molecule (*EPCAM*). Mutations in these genes are inherited in an autosomal dominant manner [10] and result in an increased risk of colon (lifetime risk 52–82%), endometrial (lifetime risk 25–60%), gastric (lifetime risk 6–13%), and ovarian (lifetime risk 4–12%) cancers [10]. Specifically, mutations in *MLH1*, *MSH2*, or *MSH6* confer risk of ovarian cancer by age 75 of 10%, 17%, or 13%, respectively [11].

Germline mutations in the mutY homolog (*MUTYH*) base excision repair gene, is best known for their role in *MUTYH*-associated polyposis (MAP), an autosomal recessive condition that confers a 63% risk of colorectal cancer by age 60 [12]. Notably, although MAP is inherited in an autosomal recessive pattern, both monoallelic and biallelic *MUTYH* carriers are at an increased risk for other cancers (Table 1), including bladder, ovarian, duodenal, breast, gastric, hepatobiliary, endometrial, and skin cancers [13,14].

The purpose of this article is to describe the role of *MUTYH* mutations in the pathogenesis of cancer and describe its emerging use in early detection, treatment decisions, and possible targeted therapies for ovarian cancer.

## 2. *MUTYH* Gene

### 2.1. MUTYH Gene Function

The *MUTYH* gene is located on the short arm of chromosome 1 (1p34.1) [15] and encodes instructions for the MYH glycosylase enzyme. This enzyme repairs DNA damage via a base excision repair mechanism. Many different carcinogens damage DNA, including reactive oxygen species, alkylating agents, DNA cross-linking agents, and radiation. Although *MUTYH* performs base excision repair and initiation of apoptosis in response to DNA damage from alkylation [16] and ultraviolet radiation [17], its primary function is to repair oxidative DNA damage [18].

Following stimulation from the *MSH6* component of the *MSH2*/*MSH6* heterodimer, *MUTYH* identifies and binds a mismatch of 8-hydroxyguanine (8-oxoG):A [19]. Myh glycosylase then excises the mismatched adenine, preventing inappropriate G:C > T:A transversions in subsequent rounds of DNA replication [15,18,20,21]. Removal of the inappropriate base-pairing generates an apurinic/apyrimidinic (AP) site [22]. A physical connection between MYH and AP endonuclease I (APE1) enables prompt action by APE1 to nick the DNA phosphodiester backbone, causing a single-strand DNA break [23] (Figure 1).

In addition to its well-known role in base excision repair, *MUTYH* also plays a role in rapid DNA damage response by inducing cell death [21,24,25]. Oka and colleagues (2008) demonstrated that single-strand mitochondrial DNA breaks performed by MYH’s base excision repair mechanism result in cell death via calpain activation [25], which is a p53 independent process [24,25]. Alternatively, nuclear DNA breaks induced by MYH glycosylase result in cell death via a poly-adenosine diphosphate (ADP) ribose polymerase (PARP) signaling pathway [25], which is mediated by the tumor suppressor protein, p53 [24] and the mismatch repair gene *MLH1* [26]. PARP senses and binds to single-strand DNA breaks, ribosylates itself and other cellular proteins, and signals apoptosis-inducing factor to translocate from the mitochondria to the nucleus, resulting in cell death [27].

Another method by which *MUTYH* responds to rapid DNA damage involves activation and phosphorylation of the tumor suppressor gene, checkpoint kinase 1 (*CHEK1*) via ataxia telangiectasia and rad3-related protein (*ATR*), another tumor suppressor gene [28,29]. When activated, *CHEK1* activates cell cycle checkpoint processes prompting DNA repair and/or apoptosis [30]. Following exposure to DNA damaging agents, Hahm and colleagues (2011) found less *CHEK1* and *ATR* phosphorylation in knockdown *MUTYH* cells than in wild-type cells [28]. When this pathway is altered, the activity of checkpoint kinase 2 (*CHEK2*) increases [28].

### 2.2. MUTYH Germline Mutations

The estimated prevalence of a heterozygous *MUTYH* germline mutation is 1–2% [31,32] and homozygous germline mutations is 0.012% [33,34]. Most pathogenic *MUTYH* mutations are missense [18,35], including Y179C (c.536A > G; previously Y165C (c.494A4G)) and G396D (c.1187G > A; previously G382D (c.1145G4A)). These mutations account for approximately 70% of all pathogenic mutations in Western populations [31,35] likely due to a genetic founder effect [21]. Although approximately 18% of all ovarian cancers are the result of an inherited predisposition [2], it is unknown how many of these ovarian cancers are directly attributable to biallelic *MUTYH* mutations. Consistent with the estimated prevalence of carrying a heterozygous *MUTYH* germline mutation [31,32], monoallelic germline *MUTYH* mutations have been identified in 1.9% of ovarian cancer patients [36].

### 2.3. MUTYH Somatic Mutations

Somatic *MUTYH* mutations are increasingly described in the literature and are noted to be present in 3.3% of all tumors curated by the Catalogue of Somatic Mutations in Cancer (COSMIC) database [37]. Similar to germline mutations, most (56.7%) mutations are missense. Over 400 different somatic *MUTYH* mutations have been curated by COSMIC and the most frequently identified is c157 + 30A > G, occurring 6% of the time [37]. Somatic *MUTYH* mutations have been described in sporadic colorectal cancer and may occur concurrent to somatic adenomatous polyposis coli (*APC*) gene mutations [38]. Furthermore, *MUTYH* somatic mutations have been described in Lynch syndrome patients and in patients whose tumors demonstrate mismatch repair deficiency but carry no mismatch repair germline mutation [39]. Outside of colon cancer, somatic *MUTYH* mutations have also been described in breast cancers [40,41] and in 0.24% of ovarian cancer samples [37].

### 2.4. MUTYH Mutation and Mechanism for Oncogenesis

While somatic and biallelic *MUTYH* mutations are associated with an increased risk of ovarian cancer, the majority of mechanistic studies have been performed in colon cancer model systems. Due to the critical role *MUTYH* plays in base excision repair, DNA with functional loss of *MUTYH* leads to an excess of 8-oxoguanine (8-oxo-G) which results in inappropriate G:C > T:A transversions in MAP colorectal tumors [18,20,42] (Figure 2). Without active *MUTYH* activity, 8-oxoguanine glycosylase (OGG1), a DNA glycosylase enzyme involved in base excision repair, may identify and remove 8-oxo-G in an inappropriate 8-oxo-G:A pairing resulting in another mechanism for incorrect G:C > T:A transversions; competition between *MUTYH* and OGG1 has demonstrated in a mouse model [21,43].

The somatic G > T transversions seen in MAP tumors are likely the result of unrepaired DNA damage from reactive oxygen species [18]. Over time, increasing numbers of G > T transversions in somatic tissues are likely to impact various oncogenes and tumor suppressor genes and lead to cancer [21,44,45]. For example, MAP colon tumors demonstrate a high prevalence of somatic *APC* [20,42] and Kirsten rat sarcoma (*KRAS*) [42] G > T transversions. MAP colon tumors have also demonstrated somatic G > T transversions in mismatch repair genes *MLH1* [46], *MSH2*, and *MSH6* [47], resulting in microsatellite instability and mimicking Lynch syndrome [46,47].

In addition to carcinogenesis resulting from failed base excision repair, tumorigenesis may result from the failure of damaged cells to undergo apoptosis [24]. Oka and colleagues (2014) demonstrated that *MUTYH* knockdown colorectal cancer cells exposed to oxidative stress failed to undergo PARP-mediated apoptosis, resulting in aberrant cell growth and risk for tumorigenesis [24]. The tumor suppressor protein, p53 [26], regulates this process: the same aberrant cell growth was identified with either p53 deficiency or *MUTYH* knockout [24]. Although no studies have investigated the effect of mutant *MUTYH* in ovarian cancer cells, the mechanism for ovarian cancer oncogenesis is likely similar to that of colorectal cancer carcinogenesis as both of these cancers are established in the MAP phenotype. Furthermore, impaired *MUTYH* function in human keratinocyte HaCaT and human embryonic kidney HEK293 cells fail to phosphorylate *ATR*/*CHEK1* [28]. Failure to prompt DNA repair and/or apoptosis [30] may shift cell checkpoint messages to *CHEK2* control. Although there is significant overlap in *CHEK1* and *CHEK2* functions [28], without *CHEK1* control, cells exposed to DNA damaging ionizing radiation fail to arrest in the G2 phase, leaving the possibility of survival and replication of damaged DNA and another potential mechanism for tumorigenesis [48]. Although these findings suggest carcinogenic plausibility, future research may test this hypothesis in ovarian cancer cell lines.

## 3. Role of MUTYH in Ovarian Cancer

### 3.1. Ovarian Cancer Risk

The genetic pathophysiology of ovarian cancer is complex. Kurman and Shih (2016) described common somatic genetic alterations driving ovarian cancer and these varied by histologic subtype [49]. For example, p53 pathway inactivation is common in high grade serous ovarian carcinomas, *KRAS* is implicated in mucinous and low grade serous ovarian carcinomas, and mismatch repair deficiency is involved in endometrioid ovarian carcinomas [49]. The *MUTYH* mutated ovarian tumors curated by COSMIC have demonstrated adenosquamous and serous histology; however, the sample size is limited [37]. Due to the multitude of DNA damage repair mechanisms controlled by *MUTYH*, it is reasonable to suspect that multiple ovarian cancer histologies may be associated with this alteration.

The risk for the development of ovarian cancer in patients with biallelic *MUTYH* germline mutations has been evaluated in two studies. In both studies, index cases were identified in patients with colon cancer who were enrolled in a cancer registry study and the association of *MUTYH* mutations with non-colonic tumors in affected family members was assessed. Among biallelic carriers, the risk of ovarian cancer was 10–14% by age 70 or 75 [13,14]. Monoallelic carriers did not demonstrate an increased risk of ovarian cancers, although numbers were relatively small and by design. all participants had a family history of colon cancer [13]. Minion and colleagues (2015) demonstrated 1.9% (9/466) of patients with ovarian cancer harbored a monoallelic *MUTYH* germline mutation [36], which is consistent with the estimated carrier rate [31,32,33,34]. It is possible that *MUTYH* carriers may progress to ovarian cancer if somatic *MUTYH* mutations co-occur, resulting in a homozygous somatic state, consistent with the well-established “two-hit hypothesis [50]. This combination of germline and somatic events was demonstrated by Thibodeau and colleagues (2019) and by Nones and colleagues (2019) in breast cancer patients [40,41]. Although COSMIC has identified somatic *MUTYH* mutations in 0.24% (3/1229) of curated ovarian cancers [37], many oncology practices only test ovarian cancer patients with next-generation sequencing for somatic *BRCA1*/*2* mutations or mismatch repair deficiency [51,52], suggesting this rate may be an underestimate.

*MUTYH* may contribute to the development of sporadic ovarian cancer via the well-described “incessant ovulation” hypothesis [53], “gonadotropin hypothesis” [54], or serous tubal intraepithelial carcinoma (STIC) hypothesis [55]. The “incessant ovulation” hypothesis suggests that more epithelial ovarian cell division during ovulation confers an increased opportunity for a somatic mutation predisposing to ovarian cancer [53,56]. *MUTYH* plays a more convincing role in the “gonadotropin hypothesis” paradigm. Pingariho and colleagues (2012) demonstrated that *MUTYH* may repair DNA damage induced by the carcinogenic metabolite glycinamide in human leukocytes [57] via base excision repair [58]. Glycinamide is the toxic metabolite of acrylamide, present in tobacco smoke and as a food and cosmetic contaminant [57]. Acrylamide is associated with increased follicle-stimulating hormone levels in premenopausal women [59], oxidative stress in rat gonads [60], and an increased risk of ovarian cancer [61]. More recently, p53 mutated lesions on fallopian tubal epithelium have been described as a precursor for many high grade serous ovarian cancers [62,63]. These STIC precancerous lesions were first described in *BRCA1*/*2* mutated patients [62] and given the interaction between p53 and *MUTYH*, carcinogenesis via this pathway seems to be a possible mechanism in *MUTYH* patients; however, additional research is needed.

### 3.2. Early Detection and Prevention of Ovarian Cancer

As ovarian cancer screening programs have not demonstrated benefit for asymptomatic women [51,64], the best option for early detection and prevention of ovarian cancer is the identification of families and patients at risk. Women who carry a biallelic germline mutation may consider similar preventative measures as patients with other mutations that confer risk. Consistent with the “incessant ovulation” hypothesis of ovarian cancer pathogenesis, a decreased risk of ovarian cancer has been demonstrated in the *BRCA1*/*2* mutated [65] and general populations [66] who have used oral contraceptives. This chemoprevention strategy may reduce the risk of ovarian cancer in populations with varying levels of pre-existing risk, including those with biallelic *MUTYH* germline mutations; however, prospective studies are needed.

The National Comprehensive Cancer Network recommends consideration of a risk-reducing salpingo-oophorectomy for patients with certain germline mutations that predispose them to ovarian cancer [67]. The recommended timing of this procedure varies and is dependent on the time in which ovarian cancer presents with that specific mutation. Data regarding the utility of this procedure for patients with germline mutations other than *BRCA1*/*2* that confer ovarian cancer risk are insufficient and require a personalized approach [67]. Data suggest a lifetime risk of ovarian cancer in biallelic *MUTYH* mutation carriers approximates 14% [13] and the median age of ovarian cancer diagnosis in this population is estimated at 51 years [14]. It seems reasonable to consider risk-reducing salpingo-oophorectomy for these patients at age 45–50. The definitive timing of this procedure must be individualized and consider each patient’s family history and risk factors. Although data are limited, patients with biallelic *MUTYH* mutations who are not yet ready for risk-reducing salpingo-oophorectomy (e.g., not yet completed childbearing or not yet ready for surgery) may undergo surveillance using a combination of serial transvaginal ultrasonography and serum cancer antigen 125 [68]. Additional studies with long-term follow-up are needed to determine the optimal surveillance protocol in this population.

### 3.3. Chemotherapeutic Considerations

Like many solid tumors, the majority of ovarian cancers require treatment with platinum-based chemotherapeutic agents [51]. The primary mechanism of action of platinum-based chemotherapeutics includes the creation of DNA-platinum adducts, leading to activation of DNA damage response system pathways, and ultimately cellular apoptosis [69,70]. Activated platinum reacts with purine DNA bases resulting in cross-linking of adjacent guanines. This DNA lesion is either repaired by one of the many DNA damage response mechanisms or deemed unrepairable prompting cellular apoptosis [70]. Although most ovarian cancer patients initially respond to platinum-taxane based chemotherapy, most recur, develop resistance, and endure a poor prognosis [51]. Platinum resistance mechanisms include: decreased intracellular accumulation, inactivation by glutathione, increased DNA repair, and failure of cells to undergo apoptosis [70].

As *MUTYH* mutations may drive some ovarian cancers, it is critical to anticipate how these tumors may respond to these therapies. Guo and colleagues (2019) demonstrated that esophageal squamous cell carcinoma cells with downregulated *MUTYH* activity contributed to cisplatin resistance via a Twist mediated epithelial-mesenchymal transition [71]. Similar to ovarian cancers, esophageal cancers are initially sensitive to upfront platinum-based chemotherapy; however, platinum resistance often ensues with continued treatment [71]. This similar treatment phenomenon suggests a similar mechanism could be involved in *MUTYH*-mutated ovarian cancers. Furthermore, tumors deficient in mismatch repair genes, including *MLH1* and *MSH2*, are less likely to recognize DNA adducts and undergo apoptosis when exposed to cisplatin [72]. Because *MUTYH*’s base excision repair mechanism functions in concert with the *MSH2*/*MSH6* heterodimer [19], tumors deficient in *MUTYH* may demonstrate similar resistance to platinum-based agents. Although the scientific community is gaining an understanding of the importance of the epithelial-mesenchymal transition and other mechanisms of platinum resistance [70,73], clinically possible methods to circumvent this problem have not been identified.

Although evidence suggests downregulated *MUTYH* activity may confer resistance to platinum-based chemotherapeutics, Jansson and coworkers (2013) demonstrated that Myh1/Rad1 homolog (Rad1) double mutant *Schizosaccharomyces pombe* (fission yeast) cells were more sensitive to cisplatin than either the Myh1 or Rad1 mutant alone [22]. Rad1, a tumor suppressor gene, is a component of the Rad9-Rad1-Hus1 complex, which senses DNA damage [74] and stimulates base excision repair [75]. Although studies investigating the response of human *MUTYH* mutant cells to various chemotherapeutic agents are lacking, the use of *S. pombe* cells may serve as a model to identify genomic markers of chemosensitivity or chemoresistance [76]. This model suggests that an isolated monoallelic *MUTYH* mutation may confer resistance to platinum-based chemotherapeutics; however, biallelic *MUTYH* mutations or a monoallelic *MUTYH* mutation that coincides with another DNA repair gene mutation may confer sensitivity. A similar phenomenon has been demonstrated in patients harboring germline *BRCA1*/*2* mutations. Ovarian cancer patients with germline *BRCA1*/*2* mutations are sensitive to platinum-based therapies and demonstrate improved survival [77]; however, if they develop a secondary somatic *BRCA1*/*2* mutation, platinum-resistance ensues [78].

Although alkylating agents are rarely used as first-line therapy for ovarian cancer, they may be used in salvage regimens [51]. Alkylating agents confer cytotoxicity by forming DNA crosslinks [69]. Fry et al. (2008) demonstrated MYH glycosylase activity in lymphoblastoid cells conferred sensitivity to alkylating agent N-methyl-N′-nitro-N-nitrosoguanidine and importantly determined that MYH knockout cells were resistant to this therapy and escaped cell death [16]. Evaluation of genomic predictors of cancer treatment efficacy and resistance often begins with lymphoblastoid cells [79]. This finding suggests that tumors with impaired *MUTYH* activity may be less responsive to alkylating chemotherapeutic agents; however, human studies are lacking to support this hypothesis.

### 3.4. Potential Targeted Therapeutics

In light of advancing knowledge about mechanisms for carcinogenesis and molecular pathways that drive the progression of cancers, significant advances have been made in developing targeted therapies for ovarian cancer. Although large clinical trials evaluating targeted *MUTYH*-therapies in ovarian cancer are unlikely, understanding the mechanisms involved in oncogenesis and successful treatments in other tumor types can help identify pathways that may be targeted with newer therapeutic agents (Figure 3).

Like other microsatellite unstable tumors, which are common in those that harbor mismatch repair mutations, MAP colon tumors display signs of an active immune response [80]. This is not altogether unexpected given the intimate association between the *MSH2*/*MSH6* heterodimer and *MUTYH*. Furthermore, MAP tumors have demonstrated somatic G > T transversions in *MSH2* and microsatellite instability [47]. The presence of tumor-infiltrating lymphocytes in ovarian cancer is reported in tumors with endometrioid histology, mismatch repair deficiency, microsatellite instability, and increased expression of program death 1 (PD-1)/programed death ligand 1 (PD-L1) [81]. Tumors that demonstrate microsatellite instability or mismatch repair deficiency may be treated with immune checkpoint (PD-1/PD-L1) inhibitors [82]; however, not all microsatellite unstable tumors are responsive to this therapy [83]. Although a *MUTYH* mutation may not currently be a specific indication for the use of PD-1/PD-L1 inhibitors, Volkov and colleagues (2020) demonstrated a pronounced tumor response in a MAP colorectal cancer patient treated with this therapy [84]. We recommend performing immunohistochemistry for mismatch repair protein expression and/or PD-1/PD-L1 expression testing on tumors of patients with *MUTYH* germline mutations as evidence suggests tumors driven by *MUTYH* mutations may be responsive to PD-1/PD-L1 inhibitors [85].

Another potential targeted therapy for ovarian cancers resulting from *MUTYH* mutations includes PARP inhibitors. Much of the research on the role of PARP inhibitors in ovarian cancer have been undertaken on patients with *BRCA1*/*2* mutations and deficiencies in the homologous recombination repair pathway [5]. *CHEK1* is a tumor suppressor involved in the homologous recombination repair pathway [86] and is activated and phosphorylated by *MUTYH* in response to DNA damage [28]. Even though the primary role of *MUTYH* is to maintain DNA integrity via base excision repair, it seems to play at least an indirect role in homologous recombination repair. *CHEK1* mutations have conferred susceptibility to PARP-inhibitors due to its role in homologous recombination repair [87]. Furthermore, results from the PRIMA/ENGOT-OV26/GOG-3012 trial demonstrated niraparib efficacy for *BRCA1*/*2* wildtype patients and patients with intact homologous recombination repair in addition to efficacy for *BRCA1*/*2* mutated and homologous recombination repair-deficient ovarian cancer patients [88]. We hypothesize the benefit of PARP inhibition may be similar to those with homologous recombination repair deficiency and should be considered in this population.

*KRAS* G:T transversions are common in MAP tumors and the success of colorectal cancer treatment options depends on *KRAS* mutational status [89,90]. Although a *KRAS* mutation confers resistance to panitumumab, a human monoclonal antibody against epidermal growth factor receptor in colorectal cancer [89], oncogenic *KRAS* mutations may be sensitive to mitogen-activated protein kinase kinase (MEK) inhibitors in other solid tumors [91], including ovarian cancer [92]. Although none of these inhibitors are currently available for use in ovarian cancer outside of a clinical trial, many are under investigation and may be options in the near future [93]. We anticipate that as we learn more about MEK inhibitor usage in ovarian cancer, somatic testing for this gene, especially in patients with *MUTYH* germline mutations, may aid treatment decisions as it does in colorectal cancer [90].

## 4. Conclusions

Biallelic *MUTYH* mutation is associated with an increased risk of ovarian cancer. Understanding the function of *MUTYH* and its associated partners is critical for determining screening, risk reduction, and therapeutic approaches for *MUTYH*-driven ovarian cancers. Much of the existing literature on *MUTYH* function comes from colorectal cancer; however, this data provides foundational information that is critical for understanding its role in ovarian cancer. Although tumors driven by *MUTYH* may be resistant to common chemotherapeutic approaches, including platinum-based agents and alkylating agents, the role of PD-1/PD-L1 inhibitors and PARP inhibitors in these tumors seems promising.

## Figures and Tables

**Figure 1 diagnostics-11-00084-f001:**
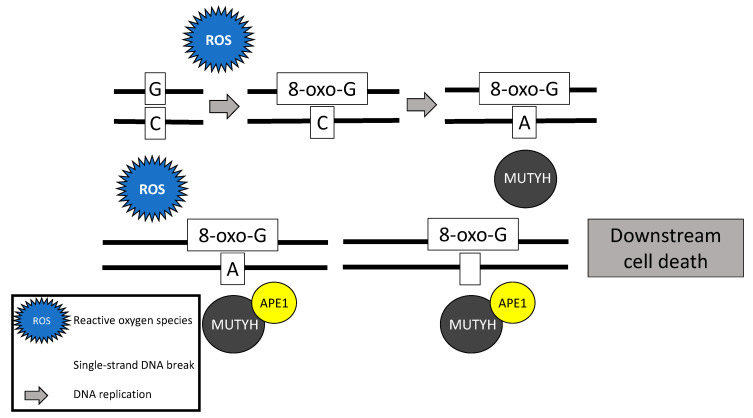
Base excision repair mechanism of the MutY homolog (*MUTYH*) in response to oxidative damage.

**Figure 2 diagnostics-11-00084-f002:**
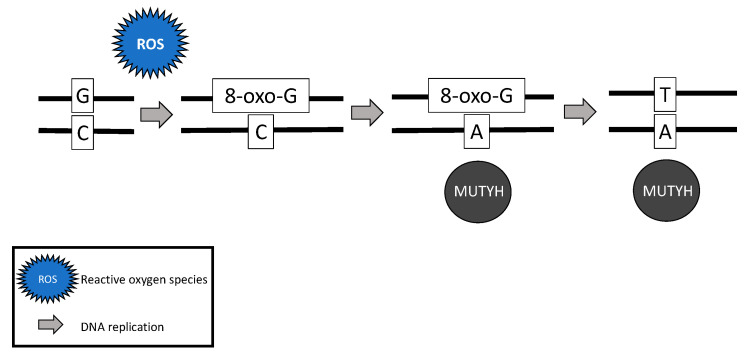
Failure of *MUTYH*’s base excision repair mechanism and resultant G > T transversion.

**Figure 3 diagnostics-11-00084-f003:**
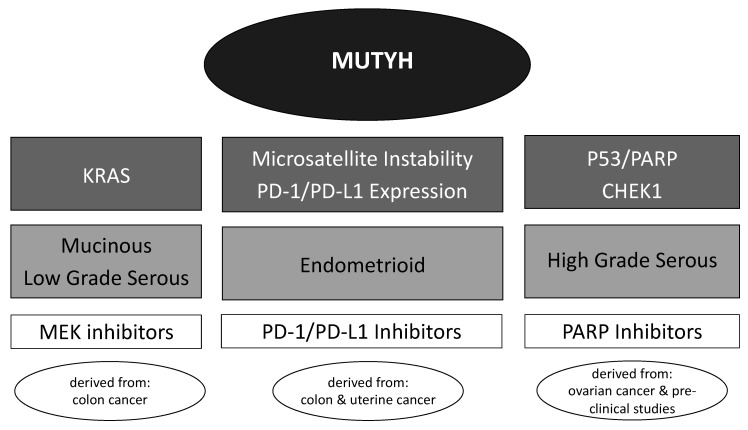
Molecular aberrations, their associated histologic subtypes, potential targeted therapeutics, and origin of treatment considerations.

**Table 1 diagnostics-11-00084-t001:** Risk of cancer phenotypes for monoallelic and biallelic *MUTYH* pathogenic mutation carriers.

Cancer Type	Risk in Monoallelic Carriers	Risk in Biallelic Carriers
Colon Cancer	Possible increased risk	63% ^1^
Bladder Cancer	Insufficient evidence	25% ^2^ (males) 8% ^2^ (females)
Ovarian Cancer	No increased risk	14% ^2^ (females)
Duodenal Cancer	Insufficient evidence	4% ^3^
Breast Cancer	11% ^1^ (females)	25% ^3^ (females)
Gastric Cancer	5% ^1^ (males) 2.3% ^1^ (females)	Insufficient evidence
Hepatobiliary Cancer	3% ^1^ (males) 1.4% ^1^ (females)	Insufficient evidence
Endometrial Cancer	3% ^1^ (females)	Possible increased risk
Skin Cancer	No increased risk	Possible increased risk

^1^ by age 60; ^2^ by age 70; ^3^ by age 75.

## Data Availability

No new data were created or analyzed in this study. Data sharing is not applicable to this article.

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
