# Peer review of "MUTYH* as an Emerging Predictive Biomarker in Ovarian Cancer"

_diagnostics, 2021, doi:10.3390/diagnostics11010084_

Round 1

Reviewer 1 Report

This is a short review which claims to encompass the role of MUTYH  gene in the development of ovarian cancer and its potential employment as a predictive and targetable biomarker of this disease. The text is well written. However, majority of the manuscript is devoted to the general information about the MUTYH gene itself, its epidemiology and associated pathophysiology, with robust accent on various mutation types and mechanisms for oncogenesisAll of that is discussed through the prism of other cancer types, not ovarian Ca. With regards to ovarian carcinomas, the authors mainly touched on the prevalence of the MUTYH mutations in patients with different OvCa histotypes, while subsequent discussion on potential early detection and chemotherapeutic recruitment of MUTYH cites studies on other cancers. Therefore, the review seems misleading. It is important that authors reorganize the text or add some type of infographics or a table summarizing studies relevant to MUTYH’s potential role as a diagnostic tool and/or therapeutic target and clearly indicate types of cancer it relates to. The summary must be useful to the research community and highlight the need for further investigation of MUTYH in ovarian cancer.  

Author Response

Thank you for your feedback. We have added additional information to better describe the existing knowledge of ovarian cancer risk in MUTYH mutation carriers; this is described in Section 3.1:  lines 165-171. We added a table to highlight the wide range of cancer phenotypes seen in those with monoallelic and biallelic mutations. You will find this on lines 62-65. In order to better contextualize how MUTYH mutations may play a role in cancer pathogenesis, we have made additions to highlight existing data from other diseases and model systems. You will find changes these changes in Section 2.4: lines 123-130 and 137-154. We have updated the manuscript to clarify the relevance of treatment considerations in other disease types and from model systems in Section 3.3: lines 229-235, 242-249, and 258-263. Furthermore, we have updated Figure 3 (lines 270-272) to delineate the origin of considerations of targeted treatments for MUTYH driven ovarian cancer.

Reviewer 2 Report

The article is well written and well structured. I especially appreciated two aspects: 1) the clarity of presentation and the organization of the paragraphs 2) the "Discussion" section: the authors in fact do not limit themselves to exposing the role of MUTYH in ovarian cancer, but also deal with its practical implications, providing examples of future correlations with chemotherapeutic agents and the new immune checkpoint inhibitors. The manuscript can be approved for publication

Author Response

We thank the reviewer for these comments and for reviewing our manuscript.

Round 2

Reviewer 1 Report

The authors have adequately addressed reviewer's concerns and considerately reorganized the text to eliminate misleading statements. The newly included table and updated treatment consideration discussion made the manuscript more meaningful in relation to ovarian cancer. I believe it is suitable for publication.